# *Rindera graeca* (Boraginaceae) Phytochemical Profile and Biological Activities

**DOI:** 10.3390/molecules25163625

**Published:** 2020-08-09

**Authors:** Christos Ganos, Nektarios Aligiannis, Ioanna Chinou, Nikolaos Naziris, Maria Chountoulesi, Tomasz Mroczek, Konstantia Graikou

**Affiliations:** 1Section of Pharmacognosy and Chemistry of Natural Products, Department of Pharmacy, National & Kapodistrian University of Athens, 15771 Zografou, Athens, Greece; chris50ganos@hotmail.com (C.G.); aligiannis@pharm.uoa.gr (N.A.); ichinou@pharm.uoa.gr (I.C.); 2Section of Pharmaceutical Technology, Department of Pharmacy, National & Kapodistrian University of Athens, 15771 Zografou, Athens, Greece; niknaz@pharm.uoa.gr (N.N.); mchountoules@pharm.uoa.gr (M.C.); 3Department of Pharmacognosy with Medicinal Plant Laboratory Unit, Medical University, ul. Chodźki 19, 20-093 Lublin, Poland; tmroczek@pharmacognosy.org

**Keywords:** phenolic compounds, quercetin-3-rutinoside-7-rhamnoside, pyrrolizidine alkaloids, antioxidant activity, HPLC-ESI-TOF-MS

## Abstract

*Rindera graeca* is a Greek endemic plant of the Boraginaceae family which has never been studied before. Consequently, this study attempted to phytochemically examine the aerial parts of this species. Nine phenolic secondary metabolites were identified, consisting of seven caffeic acid derivatives and two flavonol glucosides, namely rutin and quercetin-3-rutinoside-7-rhamnoside. These flavonoids, together with rosmarinic acid, were isolated via column chromatography and structurally determined through spectral analysis. Quercetin-3-rutinoside-7-rhamnoside is an unusual triglycoside, which is identified for the first time in *Rindera* genus and among Boraginaceae plants. This metabolite was further examined with thermal analysis and its 3D structure was simulated, revealing some intriguing information on its interaction with biological membrane models, which might have potential applications in microcirculation-related conditions. *R. graeca* was also analyzed for its pyrrolizidine alkaloids content, and it was found to contain echinatine together with echinatine N-oxide and rinderine N-oxide. Additionally, the total phenolic and flavonoid contents of *R. graeca* methanol extract were determined, along with free radical inhibition assays. High total phenolic content and almost complete inhibition at experimental doses at the free radical assays indicate a potent antioxidant profile for this plant. Overall, through phytochemical analysis and biological activity assays, insight was gained on an endemic Greek species of the little-studied *Rindera* genus, while its potential for further applications has been assessed.

## 1. Introduction

Boraginaceae is a plant family of herbs, shrubs, and trees with worldwide distribution. It includes circa 130 plant genera and approximately 2300 species [1], occurring mainly in Europe, Asia, and North America. This family is a well-known source of fatty acids, commonly present in the seeds, which have chemotaxonomic significance for Boraginaceae plants [2]. The Cynoglosseae tribe is the largest Boraginaceae subfamily consisting of around 50 genera and 900 species, spreading mainly to western Asia and the Mediterranean region [3].

The genus *Rindera* Pall. belongs to the Cynoglosseae tribe (Boraginaceae) and it includes about 25 species mostly distributed in central and eastern Europe to central Asia [4]. All of its species are perennial and adapted to dry climates. Of these, *Rindera dumanii*, *Rindera caespitosa*, and the recently identified *Rindera cetineriare* are endemic to Turkey, where plants from the *Rindera* genus are traditionally used as an anti-inflammatory medicine [5]. Essential oil from *Rindera lanata* var. *canescens*, which grows in meadows, on grasslands, and on volcanic slopes in Asia has been phytochemically analyzed and found to exert moderate in vitro antimicrobial activity [6]. Moreover, a methanolic extract of *Rindera lanata* var. *lanata* showed in vitro potential against the human rotavirus, which is responsible for gastroenteritis [5]. *Rindera umbellate*, the only endemic *Rindera* species in Serbia, appeared as a rich source of fatty acids and pyrrolizidine alkaloids (PAs) identified from aerial parts, roots, and seeds in a recent study [7]. The fruits of the widely distributed *Rindera oblongifolia* have been studied for lipids composition, showing high unsaturated fatty acid content [8]. *Rindera graeca* and *Rindera gymnandra* are the only taxa endemic to the Mediterranean region, specifically to mountainous areas of Greece and Algeria, respectively [4].

*Rindera graeca* (A.DC.) Boiss. & Heldr. is an endemic species which grows on stony Greek slopes at an elevation of 1600–2300 m, while it is listed as “Rare” in the World Conservation Monitoring Centre Plants Database [9]. The aerial parts of *R. graeca* were studied for the first time, aiming at the isolation and structural determination of its secondary metabolites and potential biological effects. Specifically, a methanolic extract from the aerial parts of the plant was researched phytochemically and evaluated for its total flavonoid content, total phenolic content, and its antioxidant activity. In the framework of our ongoing research on *R. graeca*, apart from studying the wild plant, research is being conducted on shoots from in vitro cultures and transgenic roots, which have yielded so far to the identification of caffeic acid, rosmarinic acid, lithospermic acid, and lithospermic acid B as well as rinderol, a naphthoquinone pigment [10].

PAs are a large group of alkaloids estimated to be produced by 3% of all flowering species [11]. They are based on a nitrogenous necine structure, which is biosynthesized from L-ornithine and L-arginine with putrescine and homospermidine acting as biosynthetic intermediates [12]. Moreover, pyrrolizidine alkaloid N-oxides (PANOs) are also natural plant metabolites found at nearly equal quantities to their parent alkaloid [13]. These alkaloids are a health hazard for domestic animals and humans, with neurotoxic, mutagenic, teratogenic, and carcinogenic effects. Contamination with PAs of staple foods including meat, milk, dairy products, eggs, honey, pollen products, grain, and herbal products has been reported [14]. Additionally, long-term intake of these compounds can cause hepatotoxicity and blood vessel damage since they exert their toxic effects after metabolism in the liver [15]. The Boraginaceae family [16], Cynoglossae tribe [11], and the *Rindera* genus [17] are known to be a source (in the aerial parts and roots) of PAs. Due to this fact, the aerial parts of *R. graeca* were also studied for potential PAs–PANOs content.

## 2. Results and Discussion

### 2.1. Identification and Isolation of Secondary Metabolites

The LC/MS analysis (Figure 1) yielded to the identification of nine phenolic metabolites (Table 1). Two of them are flavonol glucosides (rutin and quercetin-3-rutinoside-7-rhamnoside), together with caffeic acid and its derivatives: chlorogenic acid, rabdosiin, disodium rabdosiin salt, salvianolic acid A, rosmarinic acid, and salvianolic acid B.

Caffeic acid and chlorogenic acid as well as rutin have been identified in multiple Boraginaceae species [18]. Rosmarinic acid is commonly found in plants of the Boraginaceae family serving as a chemotaxonomic marker. This metabolite has multiple biological activities, including anti-inflammatory (lipoxygenase, cyclooxygenase inhibition), antimutagenic, antibacterial, and antiviral [19,20]. Specifically, for the latter activity, preparations including rosmarinic acid are being used for the treatment of *Herpes simplex*. Moreover, it is used widely in the cosmetics industry as an antioxidant and UVB sun radiation protector for skincare, while research is being conducted on its potential cytotoxic activity. It should be noted that rosmarinic acid is considered to be generally safe with a low toxicity profile [19,20]. Rabdosiin is a dimer of rosmarinic acid, while its disodium salt is a novel compound, recently isolated from *Alkanna sfikasiana* [1], another plant of the Boraginaceae family. It has been studied for potential cytotoxic and antiviral activities [20]. Salvianolic acid A and B are also caffeic acid derivatives, which are thought to be biosynthetically related to rosmarinic acid and are commonly found in Boraginaceae plants. The latter has shown a potent profile against microcirculation-related disorders, while its cytotoxic activity has been researched upon [20].

Furthermore, rutin, quercetin-3-rutinoside-7-rhamnoside, and rosmarinic acid were isolated, and their structures were confirmed through NMR analysis. Of these metabolites, quercetin-3-rutinoside-7-rhamnoside (Figure 2) is a little-studied compound, first isolated from *Cucumis dipsaceus* [26], isolated and identified for the first time in *Rindera* genus and the whole Boraginaceae family. As 7-substituted rutin derivatives are being used for chronic venous insufficiency (CVI) [27], a biological membrane permeation (microcirculation) related condition, the interaction of isolated quercetin-3-rutinoside-7-rhamnoside with artificial biological membrane models was assessed [28].

### 2.2. Pas Analysis Results

After purification and LC/MS analysis of the PAs extract according to the BfR method, three compounds were identified—two of PANOs and one PA (Table 2, Figure 3).

LC/MS analysis of the *R. graeca* extract was insufficient for identification, due to the fact that multiple pyrrolizidine alkaloids have the same molecular weight, but they vary in stereochemistry (and therefore retention time values). However, a comparison between received spectra of the above compounds and those of standards isolated from our scientific team from a previous study [11] suggests that the three compounds are rinderine-N-oxide, echinatine-N-oxide, and echinatine. Additionally, echinatine and its N-oxide have been identified and isolated from all previously studied *Rindera* species, namely *R. cyclodonta*, *R. echinata*, *R. baldschuanica*, *R. oblongifolia*, and *R. umbellata* [7,11,29]. Although not enough *Rindera* have been studied to establish this, the above alkaloid may actually be a chemotaxonomic marker for this genus. Additionally, rinderine also fits the profile of a possible PA in the plant, as it has been isolated from other plants of the Cynoglossae tribe to which the *Rindera* genus belongs [11,30,31].

### 2.3. Total Phenolic/Flavonoid Content and Antioxidant Activity Results

*Rindera* species have not been assayed for their phenolic and flavonoid content to the best of our knowledge. In this study, the total phenolic content (TPC) (Table 3) of the *R. graeca* methanolic extract is significantly high, at 66.5 ± 1.6 GAE mg/g compared to other Boraginaceae pants, while the total flavonoid content (TFC) (Table 3) is rather low at 9.7 ± 0.1 compared to other plants of the same family [32,33,34,35].

The DPPH and ABTS assays on the *R. graeca* methanolic extract (Table 3) showed significant inhibition of the free radicals (near 100% inhibition at higher doses for the ABTS assay) and, therefore, a significant antioxidative profile. This fact is most likely related to the above-mentioned high phenolic content, as the identified phenolic compounds are very polar containing multiple antioxidative hydroxyl groups. The mean values of the antioxidant potential of the extract in the ABTS assay were higher (99.8 ± 0.4, 96.2 ± 2.1, and 52.2 ± 1.0) compared to the DPPH assay (88.6 ± 0.2, 42.8 ± 5.3, and 24.2 ± 2.3) in all three concentrations, respectively. These results are explained by the difference in the mechanism of the radical interaction of these two assays due to the fact that ABTS acts as both a hydrophobic and hydrophilic system, while DPPH acts only as a hydrophobic one. A connection between antioxidant activity and phenolic content has been already established for plants of the Boraginaceae family [36]. It is speculated that much of the family’s biological effects (i.e., anti-inflammatory and antioxidant) are relative to the concentration of phenolic compounds, which are abundant in Boraginaceae plants [37].

### 2.4. Thermal Analysis and Molecular Dynamics Results

Quercetin-3-rutinoside-7-rhamnoside (RGMS1) was compared to rutin and quercetin standards for its ability to interact with artificial bilayer membrane models (DPPC) in two concentrations (9:0.1 and 9:0.5 lipid:flavonoid molar ratio). The evaluation was performed by thermal analysis using DSC methodology and the results are shown in Table 4. The respective thermodynamic diagrams are provided in Figure A1 in Appendix A.

Rutinoside-7-rhamnoside (RGMS1) has similar effects on the secondary transition with rutin at the lower concentration. However, at the higher concentration, it eliminates the secondary transition completely. Furthermore, it effectively halves the ΔH of the main transition, while at 9:0.5 molar ratio, it lowers the *T_onset_* of the transition by two degrees (°C).

Results from the samples containing quercetin-3-rutinoside-7-rhamnoside are indicative of interaction between the flavonoid and the nonpolar carbohydrate chains with membrane bilayer alkyl chains. It is unusual for a highly polar triglycoside to interact with the nonpolar phospholipid bilayer core. This effect is likely attributed to the conformation of the molecule, where the sugars moieties of the molecule appear to crowd together, covering the polar hydroxyl groups from the lipophilic environment, as became evident by the MD simulation snapshots (Figure 4). These effects of quercetin-3-rutinoside-7-rhamnoside in model membranes are possibly of pharmacological interest in the field of cellular membrane-related diseases, such as CVI and haemorrhoids, as is the case for other 7-substituted rutin-derived compounds [27].

## 3. Materials and Methods

### 3.1. Standards and Chemicals

The chemicals: dichloromethane (CH_2_Cl_2_), water (H_2_O), petroleum ether, acetonitrile (MeCN), ammonium formate, and methanol (ΜeOH) were of HPLC grade, purchased from Fisher Chemical (Fisher Scientific, Loughborough, Leics, UK). Formic acid was purchased from Carlo Erba Reagents. Sulfuric acid (H_2_SO_4_), benzene, gallic acid, Ehrlich’s solution, trolox, Folin & Ciocalteu’s phenol reagent, 2,2′-azino-bis(3-ethylbenzothiazoline-6-sulfonic acid) diammonium salt (ABTS), 2,2-diphenyl-1-picrylhydrazyl (DPPH), aluminum trichloride (AlCl_3_), and ethanoic anhydride (Ac_2_O) were purchased from Merck™ (Merck™, Darmstadt, Germany). Ammonium hydroxide (NH_4_OH) was purchased from Vioryl (Afidnes, Greece), Ammonia (Reag.USP. Ph. Eur. PA) from Pancreac Quimca SA and methanol-D (CD_3_OD) from Euriso-Top (Cambridge Isotope Laboratories, Tewksbury, MA, USA). Glass TLC 20 × 20 cm RP-18 F_254_ and aluminum silica gel 60 F_254_ purchased from Merck™ (Darmstadt, Germany). Rutin (Fluka™, Honeywell Specialty Chemicals, Seelze, Germany) and quercetin (Extrasynthese™, Genay, France) and sephadex LH-20 (25–100 μm, Pharmacia) were purchased.

### 3.2. Plant Material and Methanolic Extract Preparation

The aerial parts of *Rindera graeca* (Boraginaceae) were collected in 05/2014 from Mt. Parnon (Arcadia, Peloponnese, Greece). The plant material was botanically identified by Dr. E. Kalpoutzakis and a voucher specimen was deposited in the Herbarium of the Section of Pharmacognosy and Natural Product Chemistry (Dept. of Pharmacy, National and Kapodistrian University of Athens, Athens, Greece). The plant samples were naturally dried (in the shade and in a well-ventilated environment), grinded by a laboratory mill (particle size approx. 1 mm), and stored in darkness at room temperature. After that (09/2014), the dried herbal material (9 g) was macerated in MeOH. This process was thrice repeated (3 × 300 mL for 24 h each time), and after filtration and evaporation under reduced pressure, dry methanolic extract (0.67 g) was afforded.

### 3.3. Phytochemical Analysis

#### 3.3.1. Liquid Chromatography/Mass Spectrometry (LC/MS) Analysis of Phenolic Compounds

High-performance liquid chromatography-electrospray ionization-time of flight-mass spectrometry (HPLC-ESI-TOF-MS) analysis was performed by the use of an Agilent 6210 consisting of: HP 1200 chromatograph equipped with an autosampler, a binary pump, thermostated column compartment, membrane degasser, and 6210 LC/MSD mass spectrometer with a time-of-flight mass analyzer (TOF) mass analyzer (Agilent Technologies, Santa Clara, CA, USA) equipped with dual spray source: electrospray (ESI), for sample and reference masses, connected with a nitrogen (N_2_) generator (Parker Hannifin Corporation, Haverhill, MA; generating N_2_ at purities >99%), compressed air generator (Jun-Air, Oxymed, Loze, Poland), and compressed air container. The analytical column was Zorbax Stable Bond RP-18 (250 × 2.1 mm, dp = 5 μm). The column was held at 25 °C, and the mobile phase flow rate was 0.2 mL/min. As the mobile phase, the gradient of solvent mixture A (1% MeCN in H_2_O + 0.1% addition of formic acid + 10 mM ammonium formate, pH = 3.5) in B (95% MeCN in H2O + 0.1% addition of formic acid + 10 mM ammonium formate, pH = 3.5), was used as follows: 0–45 min—linear gradient: 1–60% of mixture B, 45–46 min—linear gradient: 60–90% of mixture B, 46–50 min—isocratic run 90% of mixture B. Total analysis time was 50 min, post time 15 min. The injection volume was 10 μL. Before the analysis, tuning, and calibration processes were performed using a mixture of 10 reference masses. During this process, mass measurement errors have been fixed on the level of lower than 1 ppm. Nitrogen flow was 10 L/min, gas temperature was 350 °C, and pressure was at the level of 35 psi. Analysis was performed in negative ionization mode with different fragmentation voltages (140 V, 200 V, 250 V). Mass Hunter 2.2.1 LC/MS spectra analysis software was used for the data acquisition and analysis thereof.

#### 3.3.2. Nuclear Magnetic Resonance (NMR) Analysis

^1^H-NMR spectra were obtained on a Bruker’s DRX400 spectroscopy instrument (400 MHz) using CD_3_OD as a solvent and TMS as an internal standard.

#### 3.3.3. Chromatographic Fractionation

The methanol extract (0.67 g) was subjected to molecular weight chromatography using a 33 cm column (⌀: 2.5 cm) with a Sephadex LH-20 stable phase and a 9:1 MeOH/CH_2_Cl_2_ mobile phase, yielding 15 fractions. Fraction 4 was subjected to glass silica RP-18 preparative TLC using an ΜeOH/H_2_O 50:50 solvent system, affording the phenolic metabolite quercetin-3-rutinoside-7-rhamnoside (RGMS1, 15 mg) after comparing LC/MS and NMR (Table A1) results to literature [38]. Fraction 8 yielded the pure compound quercetin-3-rutinoside (rutin) (RGMS2, 90 mg) after comparing its ^1^H-NMR spectra with literature and reference standard [24]. Fraction 12 yielded the pure compound rosmarinic acid (RGMS3, 80 mg) after comparison with literature [26].

### 3.4. Analysis of Pyrrolizidine Alkaloids (PAs)

#### 3.4.1. Preparation of PAs Extract

For the isolation and identification of PAs–PANOs from *Rindera graeca* aerial parts, the BfR method [39] was used: 2 g of dried plant material were macerated in 20 mL H_2_SO_4_ 0.05 M for 15 min under sonication. Then, the macerate was centrifuged for 10 min at 3800 rpm and the supernatant was extracted, neutralized to 7 pH and filtered, giving a PA extract.

#### 3.4.2. Isolation of PAs

For the separation and purification of PAs from *Rindera graeca* aerial parts, the Solid Phase Extraction methodology was used. A special vacuum chamber was loaded with StrataTM-X 33 μm Polymeric Reversed Phase 500 mg / 6 mL solid phase cartridges on which the mobile phase and PAs extract were added, following the specific steps of the BfR method: First, 5 mL of MeOH followed by 5 mL of H_2_O were injected so as to condition the micro-column. Then, 10 mL of the PAs extract were loaded and washed by 2 × 5 mL H_2_O injections. Subsequently, the column was dried under vacuum and finally the PAs were eluted by 2 injections of 5 mL MeOH.

The presence of PAs–PANOs was confirmed through the Mattocks/Molyneux method, first applying a 10% Ac_2_O solution in benzene/petroleum ether 4:5 and then Erlich chemical agent on a silica TLC plate, which was developed in an 88/5/2 CH_2_Cl_2_/MeOH/NH_4_OH system. The PAs were visualized as purple spots [11].

#### 3.4.3. Liquid Chromatography/Mass Spectrometry (LC/MS) Analysis of PAs

The PAs–PANOs extract was examined by qualitative LC/MS analysis. The scientific equipment used was an Agilent | 6500 Series Accurate-Mass Quadrupole Time-of-Flight (Agilent Technologies Inc., Santa Clara, CA, USA) device equipped with ESI-Jet Stream ion source and Atlantis HILIC silica column (150 × 2.1 mm, dp = 3 μm) (Waters Milform, MA, USA). The chromatograph used a diode array detector autosampler, dual grading pump, and column heater. An RP-18 stable phase was used in conjunction with a gradient elution mobile phase of 0.1% formic acid in MeOH with a stable flow of 0.25 mL/min. For this particular experiment, a positive charge ESI ion source was used [40].

### 3.5. Total Phenolic Content (TPC)

The TPC of a sample from the total methanolic extract was calculated using the Folin–Ciocalteu method. Quantification was achieved using a gallic acid reference curve. All measurements were conducted three times. Results were expressed as mg of gallic acid equivalents per gr of extract. Ultraviolet–visible spectroscopy (UV/Vis) absorption values at 765 nm were obtained using an Infinite M200 PRO TECAN reader (Tecan Group, Mannedorf, Switzerland) [41].

### 3.6. Total Flavonoid Content (TFC)

The TFC of the methanolic extract was calculated by means of an AlCl_3_ colorimetric assay. The results were obtained from a reference calibration curve of quercetin and expressed as mg of quercetin equivalents per g of extract. UV/Vis absorption values at 415 nm were obtained using an Infinite M200 PRO TECAN reader [41].

### 3.7. Antioxidant Activity

#### 3.7.1. 2,2′-Azino-Bis(3-Ethylbenzothiazoline-6-Sulfonic Acid) Diammonium Salt (ABTS) Assay

The free radical-scavenging activity of the extract was determined by ABTS radical cation (ABTS·^+^) decolorization assay [41]. The absorbance was measured at 734 nm in an Infinite M200 PRO TECAN reader and trolox was used as positive control. The radical scavenging capacity of the samples was expressed as the percentage of inhibition of the ABTS·^+^ free radical. 

#### 3.7.2. 2,2-Diphenyl-1-Picrylhydrazyl (DPPH) Assay

For this assay, discoloration of the methanolic extract solution was measured to determine the antioxidant activity thereof. The results were quantified as the percentage of inhibition of the DPPH· free radical. An Infinite M200 PRO TECAN reader provided the UV/Vis absorbance values at 517 nm [41].

### 3.8. Thermal Analysis

Quercetin-3-rutinoside-7-rhamnoside from the methanolic extract, along with flavonoid standards (rutin and quercetin) were mixed with 1,2-dipalmitoyl-sn-glycero-3-phosphocholine (DPPC, Avanti Polar Lipids) at phospholipid:flavonoid molar ratios of 9:0.1 and 9:0.5. The mixed systems were then fully hydrated in H_2_O and sealed in crucibles of 40 μL capacity. Afterwards, they were subjected to thermal analysis by utilizing the differential scanning calorimetry (DSC) method with an 822^e^ Mettler Toledo (Schwerzenbach, Switzerland) calorimeter, which was calibrated with pure indium (m = 6.2 mg), while an empty sealed crucible was used as reference. Two full cycles of heating/cooling plus one more heating scan were conducted in order to achieve optimal reproducibility of data. Temperature range was 20–60 °C and scanning rate was 5 °C/min. Before each heating cycle, samples were subjected for 10 min to a stable temperature of 20 °C to achieve system equilibrium. The second heating circle was considered for data extraction and all thermotropic parameter values (characteristic transition temperatures *T_onset,m/s_* and *T_m/s_*, enthalpy changes ΔH_m/s_, and widths at half peak height of the C_p_ profiles ΔT_1/2,m/s_) were calculated with Mettler Toledo STAR^e^ software [42]. Transition enthalpies of endothermic processes were expressed as positive values.

### 3.9. Molecular Dynamics

Flavonoid triglycoside quercetin 3-rutinoside-7-rhamnoside was inserted in DPPC phospholipid bilayers using Maestro 11 (Schrödinger) software (*Maestro-Desmond Interoperability Tools*, version 3.1; Schrodinger: New York, NY, USA, 2012) and the system was subjected to MD simulations for 50 ns using Desmond (Desmond Molecular Dynamics System, Version 3.0, D.E. Shaw Research, New York, NY, 2011. *Maestro-Desmond Interoperability Tools*, version 3.1; Schrodinger: New York, NY, USA, 2012). Complex and lipid systems were solvated using the TIP3P water model [43]. Na^+^ and Cl^−^ ions were placed in the aqueous phase to neutralize the systems and reach the experimental salt concentration of 0.150 M NaCl. Membrane generation and system solvation were conducted with the System Builder utility of Desmond. The OPLS 2005 force field [44] was used to model all protein–ligand interactions and lipids. The particle mesh Ewald method (PME) [45] was employed to calculate the long-range electrostatic interactions with a grid spacing of 0.8 Å. Van der Waals and short-range electrostatic interactions were smoothly truncated at 9.0 Å. The Nose–Hoover thermostat [46] was utilized to maintain a constant temperature in all simulations, and the Martyna–Tobias–Klein method [46] was used to control the pressure. Periodic boundary conditions were applied. The equations of motion were integrated using the multistep RESPA integrator [47] with an inner time step of 2 fs for bonded interactions and non-bonded interactions within a cutoff of 9 Å. An outer time step of 6.0 fs was used for non-bonded interactions beyond the cut-off. Each system was equilibrated using a modification of the default protocol provided in Desmond. The modification of the protocol consists of a series of restrained minimizations and MD simulations designed to relax the system, while not deviating substantially from the initial coordinates. First, two rounds of steepest descent minimization were performed using a maximum of 2000 steps and harmonic restraints of 50 kcal mol Å^−2^ applied on all solute atoms, followed by 10,000 steps of minimization without restraints. The first simulation was run for 200 ps at a temperature of 10 K in the NVT (constant number of particles, volume, and temperature) ensemble with solute heavy atoms restrained by a force constant of 50 kcal mol Å^−2^. The temperature was then raised during a 200 ps MD simulation to 310 K in the NVT ensemble with the force constant retained. The temperature of 310 K was used in MD simulations in order to ensure that the membrane state is above the main phase transition temperature of 298 K observed for DPPC bilayers [48]. The heating was then followed by equilibration simulations. First, two 1 ns stages of NPT equilibration (constant number of particles, pressure, and temperature) were performed. In the first 1 ns stage, the heavy atoms of the ligand were restrained by applying a force constant of 10 kcal mol Å^−2^ for the harmonic constraints, and in the second 1 ns stage, the heavy atoms ligand were restrained by applying a force constant of 2 kcal mol Å^−2^ to equilibrate solvent and lipids. This equilibration protocol was followed by 50 ns simulation without restraints in DPPC.

## 4. Conclusions

*Rindera graeca,* a Greek endemic plant of the Boraginaceae family, was studied phytochemically for the first time. Nine phenolic secondary metabolites were identified, consisting of seven caffeic acid derivatives, two flavonol glucosides, and three bioactive metabolites (rutin, quercetin-3-rutinoside-7-rhamnoside and rosmarinic acid) were isolated for further research. Rosmarinic acid, a metabolite with rich biological activity, is commonly found in plants of the Boraginaceae family serving as a chemotaxonomic marker. The metabolites quercetin-3-rutinoside-7-rhamnoside and rabdosiin disodium salt were identified for the first and second time in Boraginaceae family, respectively. Additionally, pyrrolizidine alkaloid content was found in *R. graeca* aerial parts and echinatine and its N-oxide, as well as rinderine N-oxide were identified. The first has been found in multiple other *Rindera* species and is possibly a chemotaxonomical marker of the genus, while rinderine-N-oxide is commonly found in Boraginaceae plants of the Cynoglossae tribe in which the *Rindera* genus belongs.

Results from free radical assays highlight a potent antioxidant profile with almost total inhibition of the radicals. This antioxidant profile is most likely related to the high total phenolic content of the plant, as phenolic compounds, such as the identified rosmarinic acid, are potent antioxidants related to the pharmacological activity of Boraginaceae plants.

Finally, the thermotropic effect of quercetin-3-rutinoside-7-rhamnoside was evaluated compared to rutin and quercetin. The results indicate interaction of the molecule with membranes on both the hydrophobic and hydrophilic regions, significantly altering their thermodynamic behavior and fluidity. This effect is likely attributed to the conformation of the molecule, similarly to other 7-substituted rutin-derived compounds, where the sugars moieties of the molecule appear to crowd together, covering the polar hydroxyl groups from the lipophilic environment.

The outcome is that quercetin-3-rutinoside-7-rhamnoside, a metabolite from *Rindera graeca* aerial parts, shows pharmacological interest in the field of cellular membrane-related diseases with potential applications in chronic venous insufficiency and hemorrhoids.

## Figures and Tables

**Figure 1 molecules-25-03625-f001:**
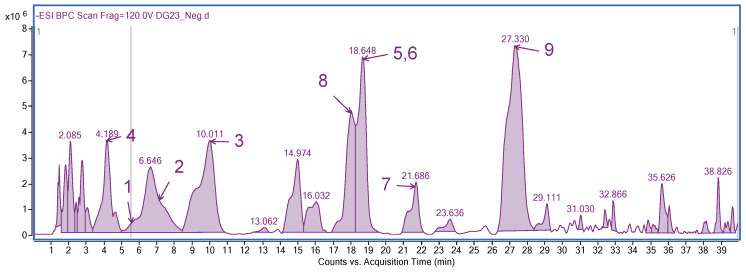
HPLC spectra of methanolic extract of *R. graeca*. **1**: Chlorogenic acid, **2**: caffeic acid, **3**: rutin, **4**: quercetin 3-rutinoside-7-rhamnoside, **5**: rabdosiin disodium salt, **6**: rabdosiin, **7**: salvianolic acid a, **8**: rosmarinic acid, and **9**: salvianolic acid B.

**Figure 2 molecules-25-03625-f002:**
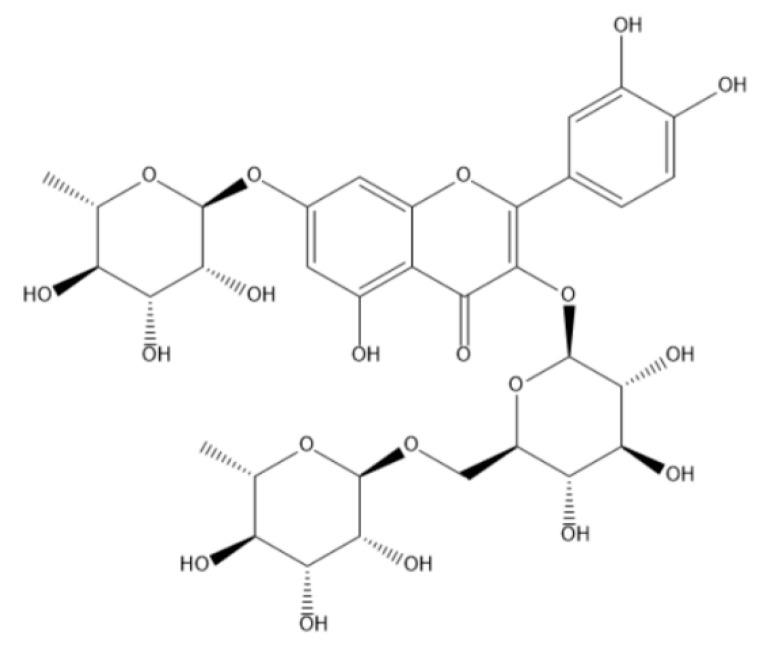
Chemical structure of quercetin-3-rutinoside-7-rhamnoside.

**Figure 3 molecules-25-03625-f003:**
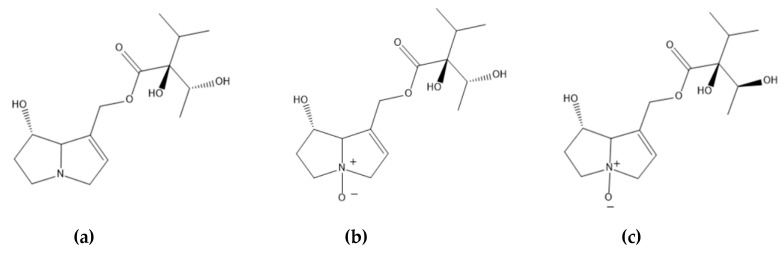
The structures of identified PA/PANOs: (**a**) echinatine, (**b**) echinatine N-oxide, (**c**) rinderine N-oxide.

**Figure 4 molecules-25-03625-f004:**
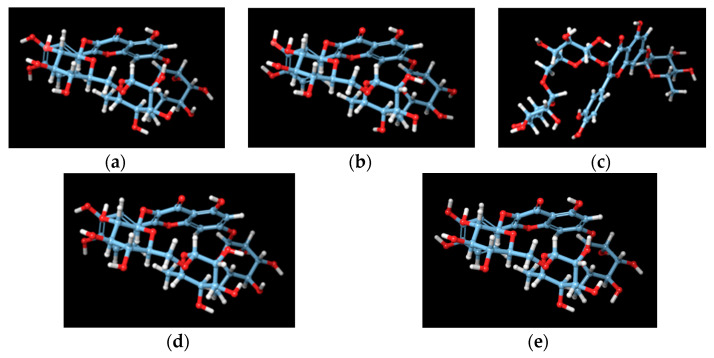
Five snapshots (**a**–**e**) from the 50-ns MD simulation of quercetin-3-rutinoside7-rhamnoside in DPPC bilayers. The four out of five 3D structures, (**a**,**b**,**d**,**e**), reveal internal wrapping of the molecule to keep the nonpolar carbohydrate chains from interaction with the alkyl chains of the bilayer.

**Table 1 molecules-25-03625-t001:** LC/MS analysis of methanol extract of *R. graeca*.

No	Rt min	Identity	Molecular Formula	[Μ − H]^−^ *m*/*z*	Ion products *m*/*z*	References
1	5.291	chlorogenic acid (3-caffeoyl quinic acid)	C_16_H_18_O_9_	353.0852	191, 173	[21]
2	7.123	Caffeic acid	C_9_H_7_O_4_	179.0326	135, 134, 89	[22]
3	10.011	Rutin (Quercetin 3-rutinoside)	C_27_H_30_O_16_	609.1445	300, 271, 301	[23]
4	4.189	Quercetin 3-rutinoside-7-rhamnoside	C_33_H_40_O_20_	755.2013	300, 609, 179	[24]
5	18.648	Disodium rabdosiin salt	C_36_H_30_O_16_Na_2_	741.1369	717, 396, 360, 161, 133	Compared to [1]
6	18.648	Rabdosiin	C_36_H_30_O_16_	717.1391	396, 360, 161, 133	[23]
7	21.385	Salvianolic acid A	C_26_H_22_O_10_	493.0985	265, 185	[22]
8	18.033	Rosmarinic acid	C_18_H_16_O_8_	359.0797	197, 179, 161, 133	[22]
9	27.330	Salvianolic acid Β	C_36_H_30_O_16_	717.1497	537, 519, 475, 339, 197	[25]

**Table 2 molecules-25-03625-t002:** LC/MS analysis of PAs extract of *R. graeca*.

Rt	Potential Identity	Chemical Formula	[M + H]^+^ *m*/*z*	Ion Products *m*/*z*
5.918	Rinderine-*N*-oxide	C_1__5_H_25_NO_7_	316.1914	172, 138, 111
6.116	Echinatine-*N*-oxide	C_1__5_H_25_NO_7_	316.1943	172, 156, 138, 111
12.800	Echinatine	C_1__5_H_25_NO_6_	300.1964	156, 138

**Table 3 molecules-25-03625-t003:** Total phenolic content (TPC), total flavonoid content (TFC), DPPH, and ABTS radical inhibition results.

**TPC GAE mg/g**	**TFC QUE mg/g**	**% DPPH Inhibition**			**% ABTS Inhibition**		
		**200 μg/mL**	**100 μg/mL**	**50 μg/mL**	**200 μg/mL**	**100 μg/mL**	**50 μg/mL**
66.5 ± 1.6	9.7 ± 0.1	88.6 ± 0.2	42.8 ± 5.3	24.2 ± 2.3	99.8 ± 0.4	96.2 ± 2.1	52.2 ± 1.0

**Table 4 molecules-25-03625-t004:** The thermodynamic parameter values of transitions observed during heating of the systems.

Sample	Molar Ratio	T_onset,m_ (°C)	T_m_ (°C)	ΔT_1/2,m_ (°C)	ΔH_m_ (Jg^−1^)	T_onset,s_ (°C)	T_s_ (°C)	ΔT_1/2,s_ (°C)	ΔH_s_ (Jg^−1^)
DPPC	-	41.05	41.88	1.45	45.01	34.56	36.54	2.10	5.08
DPPC:quercetin	9:0.1	39.59	40.79	1.43	48.47	-	-	-	-
DPPC:rutin	9:0.1	40.91	41.62	1.17	43.15	33.80	35.29	1.87	3.81
DPPC:rutin	9:0.5	41.00	41.66	0.98	48.13	33.64	35.26	1.78	4.14
DPPC:RGMS1	9:0.1	40.76	41.27	0.84	27.06	34.19	36.21	1.91	3.18
DPPC:RGMS1	9:0.5	38.23	39.59	1.42	21.76	-	-	-	-

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
