# Peer review of "Rindera graeca* (Boraginaceae) Phytochemical Profile and Biological Activities"

_molecules, 2020, doi:10.3390/molecules25163625_

Round 1
Reviewer 1 Report
The paper contained important informations and the paper could be accepted after a revision. I have some concerns and they were listed in the below.
1- Especially, plant material section. The plant was collected in 2014. At this point, when obtained the extract? r.t??. In extraction, which volume for methanol?? how long (24, 48 or 72 h)??
2- In flavonoid content, 2-10% AlCl3 ???
3- Please abbreviations should be checked again. The title 2.7.1 and 2.7.2
4- More discussion has to add in antioxidant parts.
5- Space add values and units
6- More details for biological properties of rosmarinic acid.
Author Response
According to Reviewer 1
- The authors would like to thank the reviewer for the targeted remarks
- General English language and spelling improvements have been made throughout the manuscript by a native English-speaking colleague
The paper contained important informations and the paper could be accepted after a revision. I have some concerns and they were listed in the below.
1- Especially, plant material section. The plant was collected in 2014. At this point, when obtained the extract? r.t??. In extraction, which volume for methanol?? how long (24, 48 or 72 h)??
- The date, that the extract was obtained (09/2014), was added to the paper (L 104).
- The abbreviation r.t. was changed and clarified to room temperature (L 103)
- The volume of methanol and the duration of maceration (3x300 mL, 24 h each time) was added (L105).
2- In flavonoid content, 2-10% AlCl3 ???
- In part 2.6 Total Flavonoid Content (TFC) the mistake 2-10% has been erased.
3- Please abbreviations should be checked again. The title 2.7.1 and 2.7.2
- Abbreviations were revised along the paper and they have been defined in parentheses the first time they appear in the manuscript: L. 37: approx. was changed to approximately, L. 112: the meaning of HPLC/ESI-TOF-MS was added (High - Performance Liquid Chromatography / ElectroSpray Ionization - Time of Flight - Mass Spectrometry), L. 175: the abbreviation for Quadrupole Time-of-Flight was removed as it is not used again in the paper, L. 178: the explanation for DAD (Diode Array Detector) was added, the subsections 2.7.1, 2.7.2 titles L. 199 and 206 were changed so as to explain the abbreviations of the respective free radicals, a mistake on the abbreviation for 1,2-dipalmitoyl-sn-glycero-3-phosphocholine (DPPC) was corrected on lines 233, 259, 265.
4- More discussion has to add in antioxidant parts.
- More in-depth discussion for the antioxidant results was added: “The mean values of the antioxidant potential of the extract in the ABTS assay was higher (99.8±0.4, 96.2±2.1 and 52.2±1.0) compared to the DPPH assay (88.6±0.2, 42.8±5.3 2±2.3) in all three concentrations, respectively. These results are explained by the difference in the mechanism of the radical interaction of these two assays. ABTS acts as both a hydrophobic and hydrophilic system, while DPPH acts only as hydrophobic one.” at L.339-343, along with minor changes on the same subsection.
5- Space add values and units
- Space was added between various values and their respective measurement units, all along the text. Specifically, space was added between values and min (minutes), between values and g/mg and between values and mL units.
6- More details for biological properties of rosmarinic acid.
- 277-283 a brief review of rosmarinic acid biological activities as well as toxicity was added: “This metabolite has multiple biological activities including adstringent, anti-inflammatory (lipoxygenase, cyclooxygenase inhibition), antimutagen, antibacterial and antiviral. Specifically, for the latter activity, preparations including rosmarinic acid are being used for the treatment of Herpes simplex. Moreover, is used widely in the cosmetics industry as an antioxidant and UVB sun radiaton protector for skincare, while research is being conducted on its potential cytotoxic activity. It should be noted that rosmarinic acid is considered to be generally safe, with a low toxicity profile [32,33].” Also, some information for the biological activities of rosmarinic acid derivatives, rabdosiin and salvianolic acid B in L.285-288 were added.
Reviewer 2 Report
This study examined the phytochemical composition of the the aerial parts of Rindera graeca. Nine phenolic metabolites were identified by LC/MS, this included seven caffeic acid derivatives and two flavonol glucosides (rutin and quercetin-3-rutinoside-7-rhamnoside). These flavonoids, and rosmarinic acid were isolated by column chromatography and structural determined by NMR analysis. Quercetin-3-rutinoside-7-rhamnoside was further examined with thermal analysis and its 3D-structure was simulated. Additionally, total phenolic and flavonoid content of methanolic extract were determined, as well as free radical inhibition assays.
The chemical analyses was well described and conducted. The manuscript is well written, and the information is new.
I suggest that this manuscript is therefore ready to be published as is.
The only minor comment is: Line 102: what is r.t.? room temperature ? this should be spelled in full.
Author Response
According to Reviewer 2
This study examined the phytochemical composition of the the aerial parts of Rindera graeca. Nine phenolic metabolites were identified by LC/MS, this included seven caffeic acid derivatives and two flavonol glucosides (rutin and quercetin-3-rutinoside-7-rhamnoside). These flavonoids, and rosmarinic acid were isolated by column chromatography and structural determined by NMR analysis. Quercetin-3-rutinoside-7-rhamnoside was further examined with thermal analysis and its 3D-structure was simulated. Additionally, total phenolic and flavonoid content of methanolic extract were determined, as well as free radical inhibition assays.
The chemical analyses was well described and conducted. The manuscript is well written, and the information is new.
I suggest that this manuscript is therefore ready to be published as is.
The only minor comment is: Line 102: what is r.t.? room temperature ? this should be spelled in full.
- The authors would like to thank the reviewer for the positive comments on the research
- General English language and spelling improvements have been made throughout the manuscript by a native English-speaking colleague
- The wording was changed and clarified from r.t. to room temperature along with the addition of some information on the date and details of extraction of the plant material (L.103-105).
Reviewer 3 Report
Manuscript number: molecules-891765-
Title: Rindera graeca (Boraginaceae) phytochemical profile 3 and biological activities
In the manuscript 891765 a simple study characterizing the chemical profile of the aerial parts of Rindera graeca was reported. Furthermore, the TPC, TFC and free radical inhibition assays of the obtained extract were determined. For the first time, Quercetin-3-rutinoside-7- rhamnoside was identified in Ridera genus, this compound was further examined with thermal analysis, and its 3D structure was simulated. R. graeca was also analysed for its pyrrolizidine alkaloids content and it was found to contain echinatine together with echinatine N-oxide and rinderine N-oxide.
The purpose of the manuscript is interesting because it provides information on a plant never studied and can be a starting point for further studies. This work is well presented and easy to read. Experiments were well planned and the analyses were performed by appropriate methods. The results were correctly analysed and interpreted.
It merits publication in Molecules after major revision. Detailed remarks on the text are as follows:
-Line 102: the wording 9 gr is incorrect, I think you mean 9 g. Please clarify this.
- In table 1, you change fraction m/z with ion products m/z. This wording is incorrect.
- The structure of compound 4 was confirmed through NMR analysis. For this, stereochemistry is missing in the figure1. Please clarify this.
- PAs have been identified using previously isolated reference standards. I would recommend carrying out a quantitative analysis of them. To understand the risk to which you are exposed if you want to use the plant as such.
- For a better understanding, I suggest to report the antioxidant activity as mmol Trolox equivalent (TE)/mg extract. In ABTS assay, the concentrations 200 µg/mL and 100 µg/mL showed the same % inhibition Please clarity this.
- I recommend to report a profile Full MS of methanolic extract and to highlight the peaks with the same numbering shown in table 1.
- Conclusions are missing
Author Response
According to Reviewer 3
Title: Rindera graeca (Boraginaceae) phytochemical profile 3 and biological activities
In the manuscript 891765 a simple study characterizing the chemical profile of the aerial parts of Rindera graeca was reported. Furthermore, the TPC, TFC and free radical inhibition assays of the obtained extract were determined. For the first time, Quercetin-3-rutinoside-7- rhamnoside was identified in Ridera genus, this compound was further examined with thermal analysis, and its 3D structure was simulated. R. graeca was also analysed for its pyrrolizidine alkaloids content and it was found to contain echinatine together with echinatine N-oxide and rinderine N-oxide.
The purpose of the manuscript is interesting because it provides information on a plant never studied and can be a starting point for further studies. This work is well presented and easy to read. Experiments were well planned and the analyses were performed by appropriate methods. The results were correctly analysed and interpreted.
- The authors would like to thank the reviewer for positive comments and targeted remarks on the submitted manuscript
- General English language and spelling improvements have been made throughout the manuscript by a native English-speaking colleague
It merits publication in Molecules after major revision. Detailed remarks on the text are as follows:
-Line 102: the wording 9 gr is incorrect, I think you mean 9 g. Please clarify this.
- 104: the wording was corrected from 9 gr to 9 g. Additional corrections to units around the text were conducted
- In table 1, you change fraction m/z with ion products m/z. This wording is incorrect.
- The wording was changed on Table 1 (L. 290) from Fractions to Ion Products.
- The structure of compound 4 was confirmed through NMR analysis. For this, stereochemistry is missing in the figure1. Please clarify this.
- The stereochemistry of Quercetin-3-rutinoside-7-rhamnoside was provided in the Figure 1 (line 300).
- PAs have been identified using previously isolated reference standards. I would recommend carrying out a quantitative analysis of them. To understand the risk to which you are exposed if you want to use the plant as such.
- The aim of this study was the phytochemical analysis of a plant never studied before and the evaluation of its biological activity. Furthermore, as it belongs to Boraginaceae family, known for PAs content, the BfR method was used for the confirmation of their presence and three of them were identified through LC-MS. In the future, as our team is working on many different Boraginaceae plants, we will follow this recommendation for quantitative analysis, for all of them.
- For a better understanding, I suggest to report the antioxidant activity as mmol Trolox equivalent (TE)/mg extract. In ABTS assay, the concentrations 200 µg/mL and 100 µg/mL showed the same % inhibition Please clarity this.
- Trolox was used as positive control and the activity was expressed as the percentage of inhibition of the ABTS·+ and DPPH· free radical. The experiment was made in three different concentrations (50, 100 and 200 µg/mL. The Rindera extract showed significant inhibition of the free radicals for the ABTS assay. The inhibition at the dose of 100 µg/mL is 2±2.1 and at the dose of 200 µg/mL is 99.8±0.4, which means that at a concentration higher than 100 µg/mL the extract has the highest inhibition.
- I recommend to report a profile Full MS of methanolic extract and to highlight the peaks with the same numbering shown in table 1.
- The full LC/MS spectrum of the methanolic extract was provided in the Figure A1 (Appendix) and the respective peaks, of the compounds mentioned in Table 1, were highlighted.
- Conclusions are missing
- Conclusion part was added, covering the main findings of our research (L. 390-416).
“4. Conclusions
Rindera graeca, a Greek endemic plant of the Boraginaceae family was studied phytochemically for the first time. Nine phenolic secondary metabolites were identified, consisting of seven caffeic acid derivatives and two flavonol glucosides and three bioactive metabolites (rutin, quercetin-3-rutinoside-7-rhamnoside and rosmarinic acid) were isolated for further research. Rosmarinic acid, a metabolite with rich biological activity, is commonly found in plants of the Boraginaceae family serving as a chemotaxonomic marker. The metabolites quercetin-3-rutinoside-7-rhamnoside and rabdosiin disodium salt were identified for the first and second time in Boraginaceae family, respectively. Additionally, pyrrolizidine alkaloid content was found in R. graeca aerial parts and echinatine and its N-oxide, as well as rinderine N-oxide were identified. The first has been found in multiple other Rindera species and is possibly a chemotaxonomical marker of the genus, while rinderine-N-oxide is commonly found in Boraginaceae plants of the Cynoglossae tribe in which the Rindera genus belongs. Results from free radical assays highlight a potent antioxidant profile with almost total inhibition of the radicals. This antioxidant profile is most likely related to the high total phenolic content of the plant, as phenolic compounds, such as the identified rosmarinic acid, are potent antioxidants related to the pharmacological activity of Boraginaceae plants. Finally, the thermotropic effect of quercetin-3-rutinoside-7-rhamnoside was evaluated compared to rutin and quercetin. The results indicate interaction of the molecule with membranes on both the hydrophobic and hydrophilic regions, significantly altering their thermodynamic behavior and fluidity. This effect is likely attributed to the conformation of the molecule, similarly to other 7-substituted rutin-derived compounds, where the sugars moieties of the molecule appear to crowd together, covering the polar hydroxyl groups from the lipophilic environment. The outcome of this study is that quercetin-3-rutinoside-7-rhamnoside shows pharmacological interest in the field of cellular membrane-related diseases with potential applications in chronic venous insufficiency and hemorrhoids.”
Round 2
Reviewer 3 Report
In general, the revisions requested have been accepted and clarified by the authors. The manuscript 891765 R2 merits publication in Molecules after minor revison.
I recommend to report a profile Full MS of methanolic extract and to highlight the peaks with the same numbering shown in table 1.
Author Response
- The full LC/MS spectrum of the methanolic extract was inserted in the main text as Figure 1 and the respective peaks of the compounds, mentioned in Table 1, were highlighted.